# Catalytic Acceptorless Dehydrogenation (CAD) of Secondary Benzylic Alcohols into Value-Added Ketones Using Pd(II)–NHC Complexes

**DOI:** 10.3390/molecules28134992

**Published:** 2023-06-25

**Authors:** Abeer Nasser Al-Romaizan, Manoj Kumar Gangwar, Ankit Verma, Salem M. Bawaked, Tamer S. Saleh, Rahmah H. Al-Ammari, Ray J. Butcher, Ibadur Rahman Siddiqui, Mohamed Mokhtar M. Mostafa

**Affiliations:** 1Department of Chemistry, Faculty of Science, King Abdul-Aziz University, P.O. Box 80203, Jeddah 21589, Saudi Arabia; aalromaizan@kau.edu.sa (A.N.A.-R.); smbawaked@kau.edu.sa (S.M.B.); ralammari0011@stu.kau.edu.sa (R.H.A.-A.); 2Department of Chemistry, Faculty of Science, University of Allahabad (AoU), Prayagraj 211002, Uttar Pradesh, India; mkgangwar@allduniv.ac.in (M.K.G.); ankit.chem.au@gmail.com (A.V.); irsiddiqui@allduniv.ac.in (I.R.S.); 3Department of Chemistry, College of Science, University of Jeddah, P.O. Box 80327, Jeddah 21959, Saudi Arabia; 4Department of Chemistry, Howard University, Washington, DC 20059, USA; rbutcher99@yahoo.com

**Keywords:** palladium, Pd(II)–*bis*-N-heterocyclic carbene complexes, Pd–PEPPSI complexes, secondary benzylic alcohols, value-added ketones

## Abstract

For the creation of adaptable carbonyl compounds in organic synthesis, the oxidation of alcohols is a crucial step. As a sustainable alternative to the harmful traditional oxidation processes, transition-metal catalysts have recently attracted a lot of interest in acceptorless dehydrogenation reactions of alcohols. Here, using well-defined, air-stable palladium(II)–NHC catalysts (A–F), we demonstrate an effective method for the catalytic acceptorless dehydrogenation (CAD) reaction of secondary benzylic alcohols to produce the corresponding ketones and molecular hydrogen (H_2_). Catalytic acceptorless dehydrogenation (CAD) has been successfully used to convert a variety of alcohols, including electron-rich/electron-poor aromatic secondary alcohols, heteroaromatic secondary alcohols, and aliphatic cyclic alcohols, into their corresponding value-added ketones while only releasing molecular hydrogen as a byproduct.

## 1. Introduction

The conversion of alcohols to carbonyl compounds via oxidation/dehydrogenation is one of the most important processes in organic chemistry [1,2,3] Generally, these types of reactions have been performed by using the stoichiometric portion of metal-based oxidants, such as hypochlorite salts [4], permanganate salts [5,6], chromium/pyridinium dichromate salts [7], Dess-Martin reagents [8], and Swern reagents [9], by using H_2_O_2_ [10,11,12], manganese salts [13,14], and oxygen with transition-metal catalysts [15,16]. Traditional methods have several drawbacks in terms of the environment, economy, and energy use because they typically generate a significant amount of waste or undesirable byproducts. The “green” and “sustainable” catalytic acceptorless alcohol dehydrogenation (CAAD) reactions of organic compounds, which occur in tandem with H_2_ evolution, have recently attracted attention. For acceptorless dehydrogenation (AD) reactions, no stoichiometric waste is produced because they do not require conventional oxidants or sacrificial acceptors. These processes also result in gaseous H_2_, which is valuable and might be used as an energy source [17,18]. Utilizing pricey transition metals, such as Pt, Rh, Ru, Ir, Os, Au, and Ag, numerous catalytic systems have been developed [19,20,21,22,23,24,25,26,27,28,29,30,31], as shown in Figure 1a. Despite the fact that the oxidation process itself is simple and safe, the precious metals used for the catalytic dehydrogenation of alcohols have disadvantages in terms of their toxicity and high cost. Therefore, a lot of attention has been paid recently to the development of dehydrogenation catalysts based on earth-abundant base metals, such as Mn, Fe, Co, and Ni [32,33,34,35,36,37,38,39,40]. In the field of chemical catalysis, palladium is one of the frequently studied transition metals [41,42]. Ceri Hammond and coworkers and Simon J. Freakley and coworkers recently published research on Pd-supported catalysts for acceptorless dehydrogenation (AD) reactions of various alcohols [43,44]. Numerous advantageous characteristics of the palladium catalyst, including the relatively high thermal stability of the Pd–MIC bonds, have been attributed to the concurrent emergence of 1,2,3-triazolium-derived mesoionic carbene (tz–MIC) ligands in the various cross-coupling reactions. Because these electron-rich carbenes strongly bind to the metal center, the catalyst cannot be used repeatedly without suffering a significant loss in stability. The 1,2,3–triazolium-derived mesoionic carbene (tz–MIC) palladium complexes, on the other hand, have been crucial as catalysts in homogeneous catalysis [45,46,47,48,49,50,51,52,53,54,55]; their applications in catalytic acceptorless alcohol dehydrogenation (CAAD) reactions have surprisingly remained unexplored to date, and thus we were interested in pursuing the same by using Pd–NHC metal complexes as a catalyst, as shown in Figure 1b.

## 2. Results and Discussion

### Catalytic Acceptorless Alcohol Dehydrogenation (CAAD) Using Pd(II)–NHC Catalysts (A–F)

The acceptorless dehydrogenation (AD) of different alcohols into their corresponding value-added carbonyl compounds using the catalyst palladium(II)–NHC complexes (A–F) was the main topic of this report. We started our research by screening different Pd(II)–NHC metal complexes as acceptorless dehydrogenation (AD) catalysts at 100 °C for 16 h, using **1a** as the model substrate and toluene (0.2 M) as the solvent, as part of our ongoing efforts to develop sustainable chemistry under Pd(II)–NHC catalysis as shown in Figure 1. It appears to be a very efficient catalyst among the Pd(II)–NHC (**cat D**), as evidenced by the 98% yield of the dehydrogenative product **2a** (entries 1–7, Table 1). Potassium hydroxide (KOH) and potassium tert-butoxide (KO*^t^*Bu) both produced comparable isolated yields of the dehydrogenative product **2a**, despite the fact that the various bases were tested to determine the best reaction conditions (entries 8–12). After a quick solvent screening, such as TFE (2,2,2-Trifluoroethanol), DCE (1,2-Dichloroethane), TFT (Trifluorotoluene), acetone, H_2_O (water), CH_3_CN (acetonitrile), and DMF (Dimethylformamide), the toluene appeared to be the best option (entries 13–20). When the reaction was carried out, the control experiments without a Pd–NHC catalyst produced no product, showing that the direct formation of a dehydrogenative product from alcohols under optimal conditions is essentially a catalytic process (entry 21).

Figure 2 represents the results for synthesis of 2a via different Pd catalysts

We investigated the range of substrates for CAAD (catalytic acceptorless alcohol dehydrogenation) reactions, as depicted in Figure 2. First, we investigated the extent of secondary alcohol acceptorless dehydrogenation. When the secondary alcohols had different electron-donating groups at the ortho, meta, and para positions of the arenes, they were tolerable and produced the corresponding dehydrogenated ketone products (**2b**–**e**) in good yields (68–86%, Figure 2). The reaction was unaffected by the presence of an electron-withdrawing group at the C-2, C-3, and C-4 positions of the arenes (**2f**–**k**), such as a fluoro, chloro, bromo, or nitro group, and the yields of the dehydrogenated ketone products were very good to excellent (89–94%, Figure 2). Strong yields of the dehydrogenated ketone products (88–89%) were produced by even more sterically hindered secondary alcohols with substituents at the orthopositions (**2l–m**). The oxidation of primary alcohols in the presence of minute amounts of dehydrogenated products is the subject of the following investigation (**2n–o**). To our surprise, hetero-aromatic secondary alcohols were reactive and produced the ketones (**4a–c**) in good yields of the dehydrogenated ketone products (63–68%, Figure 2) even though they frequently poisoned the catalyst through strong coordination to the metal via heteroatoms. After that, we checked challenging α,β-unsaturated cyclic aliphatic secondary alcohol (**4d**), which produced a very trace amount of product. Last but not least, we investigated the more difficult cyclic aliphatic secondary alcohols for acceptorless dehydrogenation (AD) reactions. In moderate yields (63–68%, Figure 2) of the dehydrogenated ketone products, we discovered that cyclopentanone, cyclohexanone, and cycloheptanone (**4e–g**) were produced.

A plausible reaction mechanism for the Pd–NHC complex (D)-catalyzed acceptorless dehydrogenation (AD) reaction of secondary alcohol is proposed considering the experimental findings and previously reported findings (as shown in Figure 3). The acceptorless dehydrogenation reaction’s mechanism might be comparable to some previously reported examples involving related catalytic systems [56,57,58]. First off, in the presence of KOH, the Pd–NHC catalyst D reaction with an alcohol molecule produces an intermediate called an alkoxide (I). Second, the intermediate (I) produces an aldehyde/ketone and one hydrido specie (II) by transferring the α–H to the Pd metal. The catalytic cycle is then produced by the intermediate (II) and H–Base, which evolves into H_2_ and the Pd–NHC complex (D) as shown in Figure 3.

Experiments with intermolecular hydrogen transfer were carried out to verify the H_2_ evolution in the reactions (Figure 4). In a closed system, 1-phenylethanol was hydrogenated to produce acetophenone (2a) 44% of the time and 1-(4-methoxyphenyl)ethan-1-ol (4a) 35% of the time when 1a was dehydrogenated in the presence of 1-(4-methoxyphenyl)ethan-1-one (3a) (Figure 4a). The dehydrogenation and hydrogen transfer reactions were comparatively slowed down in an open system and an Ar atmosphere (Figure 4b). A straightforward equation was used to calculate the hydrogen transfer efficiency: (yield of 4a)/(yield of 2a); 79% of the hydrogen transfer took place in a closed system (Figure 4a). In contrast, the transfer efficiency significantly dropped to 51% in an open system (Figure 4b). Additionally, we discovered H_2_ gas through gas chromatography (GC) analysis during the dehydrogenation of 1a (Appendix A) (*cf.*
Appendix A) According to the results, the catalytic reaction system operates in a dehydrogenative manner and generates H_2_ in a manner similar to other documented precious metal-based catalytic systems.

## 3. Experimental Section

### 3.1. General Considerations

Unless otherwise noted, all commercially available substances were used exactly as they were given. Toluene, ethyl acetate, and hexane were used as such from the commercial sources as reagent-grade solvents. Using CDCl_3_ solvent, ^1^H and ^13^C{^1^H} NMR measurements were taken on Bruker 400 MHz and 500 MHz spectrometers. Relative to TMS, chemical shifts (*δ*) are given in ppm, and coupling constants (*J*) are given in Hz. The chemical shifts and solvent signals that were used as references were converted to the TMS scale (CDCl_3_, *δ*C 77.0 ppm, *δ*H 7.26 ppm). Using commercial aluminum sheets precoated with silica gel, analytical thin-layer chromatography (TLC) was used to track all the reactions. Silica gel (Merck, 200−400 mesh) was used for column chromatography. Singlet (s), doublet (d), triplet (t), quartet (q), doublet of doublet (dd), doublet of triplet (dt), triplet of triplet (tt), multiplet (m), etc., are the abbreviations used for ^1^H NMR spectra to denote the signal multiplicity. The palladium(II)–NHC catalysts used in this work (**A**–**F**) were prepared according to the literature procedure [59].

### 3.2. General Procedure for the Synthesis of NHC-Pd-I_2_(pyridine) (PEPPSI) Complexes (A−C)

NHC–Pd-I_2_(pyridine) (PEPPSI) complexes (A−C) were prepared according to the modified literature procedure [59]. A mixture of triazolium iodide ligands (1.0 mmol, 1.0 equiv), PdCl_2_ (1.0 mmol, 1.0 equiv), K_2_CO_3_ (8.0 mmol, 8.0 equiv), and NaI (5.0 mmol, 5.0 equiv) was refluxed in pyridine (5 mL, 63 mmol) for 16 h. The reaction mixture was cooled to room temperature, diluted with CHCl_3_ (ca. 100 mL), and subsequently washed with saturated aqueous CuSO_4_ solution (ca. 3 × 50 mL). The organic layer was separated and dried over anhydrous Na_2_SO_4_ and filtered. The filtrate was concentrated under vacuum to give a sticky, brown residue. The residue thus obtained was further purified by column chromatography using silica gel as a stationary phase and eluted with EtOAc:petroleum ether (1:4 *v*/*v*) to give the PEPPSI complexes (A−C) as yellow solid products.

### 3.3. General Procedure for the Synthesis of Pd–bis-NHC (Cl_2_) Complexes (D−F)

Pd–bis-NHC (Cl_2_) complexes (D−F) were prepared according to the modified literature procedure [59]. A mixture of silver–NHC complexes (1.0 mmol, 1.0 equiv) and (COD)PdCl_2_ (0.50 mmol, 0.50 equiv) in CH_3_CN (ca. 50 mL) was stirred at room temperature, until the formation of an off-white AgCl precipitate were observed. The reaction mixture was filtered, and solvent was removed under vacuum to give the products (D−F) as light yellow solids.

### 3.4. General Procedure for the Preparation of Various Secondary Alcohols from Their Corresponding Ketones

All the secondary alcohols were prepared from the known procedure from the literature [60].

### 3.5. General Procedure (A) for Catalytic Acceptorless Dehydrogenation (CAD) of Secondary Alcohols into Their Corresponding Value-Added Ketones

To an oven-dried reaction tube equipped with magnetic stir bar, Pd(II)**–**NHC (cat-D), (17.2 mg, 0.025 mmol, 5 mol%), KOH (5.6 mg, 0.1 mmol, 20 mol%), and *sec*-aryl alcohol (61.1 mg, 0.5 mmol, 1 eq) were added followed by addition of 2 mL of toluene under nitrogen atmosphere. The closed reaction tube containing the reaction mixture was placed in a preheated oil bath and stirred at 100 °C for 16 h. After completion of the reaction time, the reaction mixture was cooled down to room temperature. The crude mixture was purified by flash column chromatography using silica gel as a stationary phase and hexane/ethyl acetate (95:5 *v*/*v*) as an eluent to afford the pure ketone product **2** as colorless oil in 98% (58.8 mg) yield.

### 3.6. Characterization of All Compounds

#### 3.6.1. Synthesis of Acetophenone (**2a**)

Compound **2a** was prepared according to the general procedure **A** from its corresponding secondary alcohol (61.1 mg, 0.5 mmol, 1 eq), and the crude product was further purified by column chromatography using silica as a stationary phase and eluting with petroleum ether/EtOAc (*v*/*v* 95:5) to afford **2a** as colorless oil in 98% (60.8 mg) yield. The NMR data of **2a** are in accordance with the literature [61].



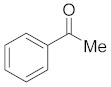



^1^H NMR (400 MHz, CDCl_3_): *δ* 7.95−7.90 (m, 2H), 7.51 (d, *J* = 8.0 Hz, 1H), 7.43 (t, *J* = 7.3 Hz, 2H), 2.57 (s, 3H). ^13^C NMR (100 MHz, CDCl_3_): *δ* 198.2, 137.2, 133.2, 128.6, 128.4, 26.6.

#### 3.6.2. Synthesis of 1-p-Tolylethanone (**2b**)

Compound **2b** was prepared according to the general procedure **A** from its corresponding secondary alcohol (68.1 mg, 0.5 mmol, 1 eq), and the crude product was further purified by column chromatography using silica as a stationary phase and eluting with petroleum ether/EtOAc (*v*/*v* 95:5) to afford **2b** as colorless oil in 86% (59.4 mg) yield. The NMR data of **2b** are in accordance with the literature [61].



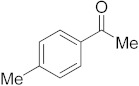



^1^H NMR (400 MHz, CDCl_3_): *δ* 7.83–7.81 (m, 2H), 7.21 (d, *J* = 8.0 Hz, 2H), 2.53 (s, 3H), 2.36 (s, 3H). ^13^C NMR (100 MHz, CDCl_3_): *δ* 197.9, 143.9, 134.7, 129.3, 128.5, 26.6, 21.7.

#### 3.6.3. Synthesis of 1-(3-Methoxyphenyl)ethanone (**2c**)

Compound **2c** was prepared according to the general procedure **A** from its corresponding secondary alcohol (76.1 mg, 0.5 mmol, 1 eq), and the crude product was further purified by column chromatography using silica as a stationary phase and eluting with petroleum ether/EtOAc (*v*/*v* 95:5) to afford **2c** as colorless oil in 78% (60.1 mg) yield. The NMR data of **2c** are in accordance with the literature [61].



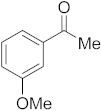



^1^H NMR (400 MHz, CDCl_3_): *δ* 7.50–7.48 (m, 1H), 7.45–7.44 (m, 1H), 7.32 (t, *J* = 8.0 Hz, 1H), 7.10–7.05 (m, 1H), 3.80 (s, 3H), 2.55 (s, 3H). ^13^C NMR (100 MHz, CDCl_3_): *δ* 197.9, 159.7, 138.4, 129.5, 121.1, 119.5, 112.3, 55.3, 26.7.

#### 3.6.4. Synthesis of 1-(4-Methoxyphenyl)ethanone (**2d**)

Compound **2d** was prepared according to the general procedure **A** from its corresponding secondary alcohol (76.1 mg, 0.5 mmol, 1 eq), and the crude product was further purified by column chromatography using silica as a stationary phase and eluting with petroleum ether/EtOAc (*v*/*v* 95:5) to afford **2d** as colorless oil in 76% (57.0 mg) yield. The NMR data of **2d** are in accordance with the literature [61].



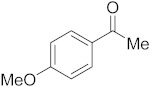



^1^H NMR (400 MHz, CDCl_3_): *δ* 7.95–7.90 (m, 2H), 6.94–6.90 (m, 2H), 3.86 (s, 3H), 2.55 (s, 3H). ^13^C NMR (100 MHz, CDCl_3_): *δ* 196.8, 163.6, 130.7, 130.4, 113.8, 55.6, 26.4.

#### 3.6.5. Synthesis of 1-(4-Aminophenyl)ethanone (**2e**)

Compound **2e** was prepared according to the general procedure **A** from its corresponding secondary alcohol (68.5 mg, 0.5 mmol, 1 eq), and the crude product was further purified by column chromatography using silica as a stationary phase and eluting with petroleum ether/EtOAc (*v*/*v* 95:5) to afford **2e** as yellow solid in 68% (47.2 mg) yield. The NMR data of **2e** are in accordance with the literature [62].



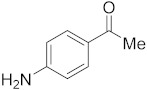



^1^H NMR (400 MHz, CDCl_3_): *δ* 7.78 (dd, *J* = 8.6 Hz, *J* = 1.5 Hz, 2H), 6.62 (dd, *J* = 8.6 Hz, *J* = 1.3 Hz, 2H), 4.20 (s, 2H), 2.50 (s, 3H). ^13^C NMR (100 MHz, CDCl_3_): *δ* 196.7, 151.4, 130.9, 127.9, 113.8, 26.2.

#### 3.6.6. Synthesis of 1-(4-Fluorophenyl)ethanone (**2f**)

Compound **2f** was prepared according to the general procedure **A** from its corresponding secondary alcohol (77.0 mg, 0.5 mmol, 1 eq), and the crude product was further purified by column chromatography using silica as a stationary phase and eluting with petroleum ether/EtOAc (*v*/*v* 97:3) to afford **2f** as colorless oil in 94% (73.4 mg) yield. The NMR data of **2f** are in accordance with the literature [61].



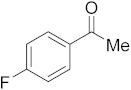



^1^H NMR (400 MHz, CDCl_3_): *δ* 7.92–7.90 (m, 2H), 7.05 (t, *J* = 8.7 Hz, 2H), 2.51 (s, 3H). ^13^C NMR (100 MHz, CDCl_3_): *δ* 196.4, 167.0, 164.5, 133.6, 133.5, 131.0, 130.9, 115.7, 115.5, 26.5. ^19^F NMR (377 MHz, CDCl_3_): *δ* −105.5.

#### 3.6.7. Synthesis of 1-(3-Chlorophenyl)ethanone (**2g**)

Compound **2g** was prepared according to the general procedure **A** from its corresponding secondary alcohol (78.3 mg, 0.5 mmol, 1 eq), and the crude product was further purified by column chromatography using silica as a stationary phase and eluting with petroleum ether/EtOAc (*v*/*v* 97:3) to afford **2g** as colorless oil in 90% (71.3 mg) yield. The NMR data of **2g** are in accordance with the literature [62].



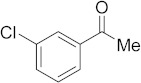



^1^H NMR (400 MHz, CDCl_3_): *δ* 7.89 (d, *J* = 1.6 Hz, 1H), 7.80–7.78 (m, 1H), 7.51–7.40 (m, 1H), 7.35 (t, *J* = 8.0 Hz, 1H), 2.56 (s, 3H). ^13^C NMR (100 MHz, CDCl_3_): *δ* 196.7, 138.6, 134.9, 133.0, 130.0, 128.4, 126.5, 26.7.

#### 3.6.8. Synthesis of 1-(4-Chlorophenyl)ethanone (**2h**)

Compound **2h** was prepared according to the general procedure **A** from its corresponding secondary alcohol (78.0 mg, 0.5 mmol, 1 eq), and the crude product was further purified by column chromatography using silica as a stationary phase and eluting with petroleum ether/EtOAc (*v*/*v* 97:3) to afford **2h** as yellow liquid in 89% (70.5 mg) yield. The NMR data of **2h** are in accordance with the literature [62].



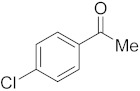



^1^H NMR (400 MHz, CDCl_3_): *δ* 7.82–7.80 (m, 2H), 7.36–7.33 (m, 2H), 2.51 (s, 3H). ^13^C NMR (100 MHz, CDCl_3_): *δ* 196.7, 139.5, 135.5, 129.7, 128.9, 26.5.

#### 3.6.9. Synthesis of 1-(2-Bromophenyl)ethanone (**2i**)

Compound **2i** was prepared according to the general procedure **A** from its corresponding secondary alcohol (100 mg, 0.5 mmol, 1 eq), and the crude product was further purified by column chromatography using silica as a stationary phase and eluting with petroleum ether/EtOAc (*v*/*v* 97:3) to afford **2i** as colorless oil in 92% (92.9 mg) yield. The NMR data of **2i** are in accordance with the literature [61].



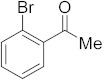



^1^H NMR (400 MHz, CDCl_3_): *δ* 7.56 (dd, *J* = 8 Hz, *J* = 1.0 Hz, 1H), 7.42 (dd, *J* = 7.7 Hz, *J* = 1.8 Hz, 1H), 7.34–7.30 (m, 1H), 7.25 (td, *J* = 7.6 Hz, *J* = 1.8 Hz, 1H), 2.60 (s, 3H). ^13^C NMR (100 MHz, CDCl_3_): *δ* 199.9, 145.9, 137.9, 134.4, 130.8, 127.5, 124.4, 128.9, 30.2.

#### 3.6.10. Synthesis of 1-(4-Bromophenyl)ethanone (**2j**)

Compound **2j** was prepared according to the general procedure **A** from its corresponding secondary alcohol (100 mg, 0.5 mmol, 1 eq), and the crude product was further purified by column chromatography using silica as a stationary phase and eluting with petroleum ether/EtOAc (*v*/*v* 97:3) to afford **2j** as colorless oil in 91% (91.9 mg) yield. The NMR data of **2j** are in accordance with the literature [61].



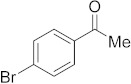



^1^H NMR (400 MHz, CDCl_3_): *δ* 7.83–7.80 (m, 2H), 7.63–7.59 (m, 2H), 2.59 (s, 3H). **^13^C** NMR (100 MHz, CDCl_3_): *δ* 197.1, 136.0, 132.0, 129.9, 128.4, 26.6.

#### 3.6.11. Synthesis of 1-(2-Nitrophenyl)ethanone (**2k**)

Compound **2k** was prepared according to the general procedure **A** from its corresponding secondary alcohol (83.5 mg, 0.5 mmol, 1 eq), and the crude product was further purified by column chromatography using silica as a stationary phase and eluting with petroleum ether/EtOAc (*v*/*v* 97:3) to afford **2k** as yellow liquid in 93% (78.5 mg) yield. The NMR data of **2k** are in accordance with the literature [63].



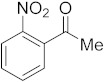



^1^H NMR (500 MHz, CDCl_3_): *δ* 8.02 (dd, *J* = 8.2, *J* = 0.9 Hz, 1H), 7.68 (td, *J* = 7.5, *J* = 1.1 Hz, 1H), 7.58–7.54 (m, 1H), 7.41 (dd, *J* = 7.6, *J* = 1.3 Hz, 1H), 2.52 (s, 3H). ^13^C NMR (100 MHz, CDCl_3_): *δ* 199.8, 145.9, 137.9, 134.3, 130.8, 127.4, 124.4, 30.1.

#### 3.6.12. Synthesis of 1-(Naphthalen-1-yl)ethanone (**2l**)

Compound **2l** was prepared according to the general procedure **A** from its corresponding secondary alcohol (86.1 mg, 0.5 mmol, 1 eq), and the reaction mixture was purified by flash column chromatography (5% EtOAc/Hexane) to afford **2l** as white solid in 88% (76.6 mg) yield. The NMR data of **2l** are in accordance with the literature [64].



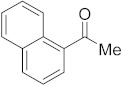



^1^H NMR (400 MHz, CDCl_3_): *δ* 8.76 (d, *J* = 8.6 Hz, 1H), 8.00–7.86 (m, 3H), 7.63–7.47 (m, 3H), 2.74 (s, 3H). ^13^C NMR (100 MHz, CDCl_3_): *δ* 202.0, 135.5, 134.1, 133.2, 130.2, 128.8, 128.5, 128.2, 126.5, 126.1, 124.4, 30.1.

#### 3.6.13. Synthesis of 1-(Naphthalen-2-yl)ethanone (**2m**)

Compound **2m** was prepared according to the general procedure **A** from its corresponding secondary alcohol (86.1 mg, 0.5 mmol, 1 eq), and the reaction mixture was purified by flash column chromatography (5% EtOAc/Hexane) to afford **2m** as colorless oil in 89% (77.5 mg) yield. The NMR data of **2m** are in accordance with the literature [62].



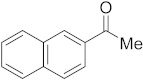



^1^H NMR (400 MHz, CDCl_3_): *δ* 8.41 (s, 1H), 8.01 (dd, *J* = 8.7, *J* = 1.7 Hz, 1H), 7.91 (d, *J* = 7.9 Hz, 1H), 7.84–7.82 (m, 2H), 7.58–7.50 (m, 2H), 2.67 (s, 3H). ^13^C NMR (100 MHz, CDCl_3_): *δ* 198.0, 135.6, 134.4, 132.5, 130.2, 129.5, 128.5, 128.4, 127.8, 126.8, 123.9, 26.7.

#### 3.6.14. Synthesis of 1-(Pyridin-2-yl)ethanone (**4a**)

Compound **4a** was prepared according to the general procedure **A** from its corresponding secondary alcohol (61.5 mg, 0.5 mmol, 1 eq), and the reaction mixture was purified by flash column chromatography (5% EtOAc/Hexane) to afford **4a** as colorless oil in 68% (42.5 mg) yield. The NMR data of **4a** are in accordance with the literature [63].



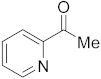



^1^H NMR (400 MHz, CDCl_3_): *δ* 8.61 (d, *J* = 4.8 Hz, 1H), 7.97 (d, *J* = 7.8 Hz, 1H), 7.77 (td, *J* = 7.8, *J* = 1.7 Hz, 1H), 7.42–7.37 (m, 1H), 2.66 (s, 3H). ^13^C NMR (100 MHz, CDCl_3_): *δ* 201.1, 153.6, 149.0, 136.9, 127.2, 121.7, 25.8.

#### 3.6.15. Synthesis of 1-(Pyridin-3-yl)ethanone (**4b**)

Compound **4b** was prepared according to the general procedure **A** from its corresponding secondary alcohol (61.5 mg, 0.5 mmol, 1 eq), and the reaction mixture was purified by flash column chromatography (5% EtOAc/Hexane) to afford **4b** as colorless oil in 65% (40.6 mg) yield. The NMR data of **4b** are in accordance with the literature [61,63].



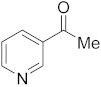



^1^H NMR (400 MHz, CDCl_3_): *δ* 9.05 (d, *J* = 1.9 Hz, 1H), 8.67 (dd, *J* = 4.8, *J* = 1.6 Hz, 1H), 8.14 (dt, *J* = 7.9, *J* = 1.9 Hz, 1H), 7.33 (dd, *J* = 7.9, *J* = 4.9 Hz, 1H), 2.54 (s, 3H). ^13^C NMR (100 MHz, CDCl_3_): *δ* 196.8, 153.5, 149.9, 135.5, 132.2, 123.6, 26.7.

#### 3.6.16. Synthesis of 1-(Thiophen-2-yl)ethanone (**4c**)

Compound **4c** was prepared according to the general procedure **A** from its corresponding secondary alcohol (64.1 mg, 0.5 mmol, 1 eq), and the crude product was further purified by column chromatography using silica as a stationary phase and eluting with petroleum ether/EtOAc (*v*/*v* 95:5) to afford **4c** as colorless oil in 63% (41.0 mg) yield. The NMR data of **4c** are in accordance with the literature [61].



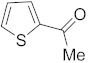



^1^H NMR (400 MHz, CDCl_3_): *δ* 7.66 (dd, *J* = 3.7 Hz, *J* = 1.0 Hz, 1H), 7.62 (dd, *J* = 4.9 Hz, *J* = 1.0 Hz, 1H), 7.10 (dd, *J* = 4.8 Hz, *J* = 3.8 Hz, 1H), 2.54 (s, 3H). ^13^C NMR (100 MHz, CDCl_3_): *δ* 190.8, 144.6, 133.9, 132.6, 128.2, 26.9.

#### 3.6.17. Synthesis of Cyclopentanone (**4e**)

Compound **4e** was prepared according to the general procedure **A** from its corresponding secondary alcohol (43.0 mg, 0.5 mmol, 1 eq), and the crude product was further purified by column chromatography using silica as a stationary phase and eluting with petroleum ether/EtOAc (*v*/*v* 95:5) to afford **4e** as colorless oil in 39% (17.1 mg) yield. The NMR data of **4e** are in accordance with the literature [65].



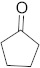



^1^H NMR (500 MHz, CDCl_3_): *δ* 2.10–1.92 (m, 4H), 1.91–1.85 (m, 4H). ^13^C NMR (125 MHz, CDCl_3_): *δ* 220.7, 38.3, 23.2.

#### 3.6.18. Synthesis of Cyclohexanone (**4f**)

Compound **4f** was prepared according to the general procedure **A** from its corresponding secondary alcohol (50.0 mg, 0.5 mmol, 1 eq), and the crude product was further purified by column chromatography using silica as a stationary phase and eluting with petroleum ether/EtOAc (*v*/*v* 95:5) to afford **4f** as colorless oil in 43% (21.9 mg) yield. The NMR data of **4f** are in accordance with the literature [61,65].



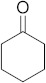



^1^H NMR (500 MHz, CDCl_3_): *δ* 2.22 (t, *J* = 6.7 Hz, 4H), 1.78–1.73 (m, 4H), 1.64–1.57 (m, 2H). ^13^C NMR (125 MHz, CDCl_3_): *δ* 211.8, 41.9, 26.9, 24.9.

#### 3.6.19. Synthesis of Cycloheptanone (**4g**)

Compound **4g** was prepared according to the general procedure **A** from its corresponding secondary alcohol (57.0 mg, 0.5 mmol, 1 eq), and the crude product was further purified by column chromatography using silica as a stationary phase and eluting with petroleum ether/EtOAc (*v*/*v* 95:5) to afford **4g** as colorless oil in 48% (27.8 mg) yield. The NMR data of **4g** are in accordance with the literature [65].



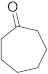



^1^H NMR (500 MHz, CDCl_3_): *δ* 2.42–2.40 (m, 4H), 1.64–1.58 (m, 8H). ^13^C NMR (125 MHz, CDCl_3_): *δ* 215.3, 43.9, 30.4, 24.4.

### 3.7. Mercury Drop Experiment Performed at Varying Time Intervals

#### 3.7.1. Mercury Addition at the Start of the Reaction

A 10 mL vial was charged with a mixture of 1-phenylethan-1-ol (61.1 mg, 0.5 mmol, 1 eq) and KOH (5.6 mg, 0.1 mmol, 20 mol%) in molar ratio of 5:1, and mercury (0.121 g, 0.603 mmol) was added subsequently. The palladium complex D (17.2 mg, 0.025 mmol, 5 mol%) was added to the mixture, followed by toluene (ca. 2 mL) solvent, and closed reaction tube containing the reaction mixture was placed in a preheated oil bath and stirred at 100 °C for 6 h. The reaction mixture was cooled to room temperature, and water (ca. 12 mL) was added. The resulting mixture was extracted with EtOAc (ca. 50 mL). The water layer was further extracted with EtOAc (ca. 3 × 20 mL). The crude mixture was purified by flash column chromatography using silica gel as a stationary phase and hexane/ethyl acetate (95:5 *v*/*v*) as an eluent to afford the pure ketone product **2a** as colorless oil in 84% (52.1 mg) yield.

#### 3.7.2. Mercury Addition after 2 h of Reaction Time 

A 10 mL vial was charged with a mixture of 1-phenylethan-1-ol (61.1 mg, 0.5 mmol, 1 eq) and KOH (5.6 mg, 0.1 mmol, 20 mol%) in molar ratio of 5:1. The palladium complex-D (17.2 mg, 0.025 mmol, 5 mol%) was added to the mixture, followed by toluene (ca. 2 mL), and then the reaction mixture was heated at 100 °C for 2 h. Mercury (0.126 g, 0.628 mmol) was added, and the reaction mixture was further heated at 100 °C for 4 h. The reaction mixture was cooled to room temperature, and water (ca. 12 mL) was added. The resulting mixture was extracted with EtOAc (ca. 50 mL). The water layer was further extracted with EtOAc (ca. 3 × 20 mL). The crude mixture was purified by flash column chromatography using silica gel as a stationary phase and hexane/ethyl acetate (95:5 *v*/*v*) as an eluent to afford the pure ketone product **2a** as colorless oil in 72% (44.7 mg) yield.

### 3.8. Experimental Procedure for Detection of H_2_ Gas by GC 

A 250 mL oven-dried Schlenk tube with a rubber septum was filled with 1-phenylethanol (10 mmol, 1.21 mL), palladium complex D (17.2 mg, 0.025 mmol, 5 mol%), KOH (5.6 mg, 0.1 mmol, 20 mol%), and toluene (10 mL). This was carried out in an atmosphere of argon. The reaction medium was stirred at 100 °C in a closed environment for 24 h. With the help of a Hamilton syringe, gas that had been filled to the top of a Schlenk tube was sampled in order to detect hydrogen.

## 4. Conclusions

In this report, we demonstrate an efficient protocol for the catalytic acceptorless dehydrogenation (CAD) reaction of secondary alcohols into their corresponding value-added ketones and the release of molecular hydrogen (H_2_) as the sole side product by using well-defined, air-stable, Pd(II)–NHC catalysts under mild reaction conditions. This protocol is a highly efficient, an economical, and an environmentally friendly alternative to all other methods, which require harsh reaction conditions/hazardous solvents and reagents.

## Data Availability

The catalytic product NMR data are contained within the supporting information of this article. Other data have been cited and listed in the bibliography.

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
