# Peer review of "Catalytic Acceptorless Dehydrogenation (CAD) of Secondary Benzylic Alcohols into Value-Added Ketones Using Pd(II)–NHC Complexes"

_molecules, 2023, doi:10.3390/molecules28134992_

Round 1
Reviewer 1 Report
This manuscript describes the catalytic properties of palladium complexes in acceptorless dehydrogenation of secondary alcohols. The palladium complexes are interesting catalytic systems although have been previously described which limits its novelty of the manuscript. In the end, we have a manuscript focused in acceptorless dehydrogenation properties. The optimization of reaction conditions and the scope of the reaction is well-described. However, the authors do not provide any mechanistic studies for claiming a bifunctional reaction mechanism. In addition, the proposed mechanism is a traditional inner-sphere where any ligand plays a role in the transformation. I do consider that the acceptorless dehydrogenation of alcohols using palladium complexes is a hot topic in research, however I can not support the publication of the present manuscript in its actual form.
Minor points:
-The authors claim that palladium complexes have been described previously in ref. 59, but such article does not contain such information.
-The authors should include at least a graphical representation of yield/conversion vs. time. This provides useful information about apparent reaction kinetics.
-Palladium complexes have a great propensity to form heterogeneous species. Please check if this is a potential alternative to the proposed mechanism. There are different experiments that should be conducted such as mercury test, hot filtration experiments, color change of catalytic reactions, etc.
-At least one 1H NMR of reaction crude without purification should be included.
-The NMR spectra included in the supporting information should include a brief statement of the procedure use for characterization indicating the purification process.
Author Response
Manuscript ID: molecules-2433079R1 Date: June, 10th, 2023
Response to Reviewers
Dear Editor,
Thank you for giving us the opportunity to submit a revised version of draft of the manuscript entitled “Catalytic Acceptorless Dehydrogenation (CAD) of Secondary Benzylic Alcohols into Value-added Ketones using Pd(II)-NHC Complexes” by Professor Saleh et al. to be considered for publication as a research article in your prestigious journal of molecules - MDPI. We appreciate the time and effort that you and the reviewers dedicated to providing feedback on our manuscript and are grateful for the insightful comments on and valuable improvements to our paper. We have incorporated most of the suggestions made by the reviewers. Those changes are highlighted within the revised manuscript. Please see below, in blue, for a point-by-point response to the reviewers’ comments and concerns. All the reference has been rechecked and revised as suggested by the reviewers.
Reviewer’s comments & responses
Recommendation: major revision.
This manuscript describes the catalytic properties of palladium complexes in acceptorless dehydrogenation of secondary alcohols. The palladium complexes are interesting catalytic systems although have been previously described which limits its novelty of the manuscript. In the end, we have a manuscript focused in acceptorless dehydrogenation properties. The optimization of reaction conditions and the scope of the reaction is well-described. However,
- the authors do not provide any mechanistic studies for claiming a bifunctional reaction mechanism. In addition, the proposed mechanism is a traditional inner-sphere where any ligand plays a role in the transformation. I do consider that the acceptorless dehydrogenation of alcohols using palladium complexes is a hot topic in research; however I cannot support the publication of the present manuscript in its actual form.
Response: Thank you very much for your valuable comment which really add to quality of manuscript. We add in our revised version a mechanistic study that prove the bifunctional reaction mechanism as
Experiments with intermolecular hydrogen transfer were carried out to verify H2 evolution in the reactions (Scheme 4). In a closed system, 1-phenylethanol was hydrogenated to produce acetophenone (2a) in 44% of the time and 1-(4-methoxyphenyl)ethan-1-ol (4a) in 35% of the time when 1a was dehydrogenated in the presence of 1-(4-methoxyphenyl)ethan-1-one (3a) (Scheme 4a). The dehydrogenation and hydrogen transfer reactions were comparatively slowed down in an open system and an Ar atmosphere (Scheme 4b). A straightforward equation was used to calculate the hydrogen transfer efficiency: [yield of 4a]/[yield of 2a]; 79% of hydrogen transfer took place in a closed system (Scheme 4a). In contrast, the transfer efficiency significantly dropped to 51% in an open system (Scheme 4b). Additionally, we discovered H2 gas through gas chromatography (GC) analysis during the dehydrogenation of 1a (Figure S1’’). According to the results, the catalytic reaction system operates in a dehydrogenative manner and generates H2 in a manner similar to other documented precious metal-based catalytic systems.
“Scheme 4” Experiments with intermolecular hydrogen transfer were carried out to verify H2 evolution in the reactions
- The authors claim that palladium complexes have been described previously in ref. 59, but such article does not contain such information.
Response: Thank you so much for pointing this out. now we have revised the reference [59]. As suggested by reviewer-1.
- The authors should include at least a graphical representation of yield/conversion vs. time. This provides useful information about apparent reaction kinetics.
Response: Thank you very much for your valuable comment which really add to quality of manuscript. We add in our revised version the requested graph.
- Palladium complexes have a great propensity to form heterogeneous species. Please check if this is a potential alternative to the proposed mechanism. There are different experiments that should be conducted such as mercury test, hot filtration experiments, color change of catalytic reactions, etc.
Response: Thank you for pointing this out. and here we have included the mercury drop experiment was performed at varying time intervals, in the revised manuscript, as suggested by reviewer-1.
- At least one 1H NMR of reaction crude without purification should be included
Response:
Thank you for pointing this out. The reviewer's comment is accepted; and we have included one crude 1H NMR spectra of the product without purification (Figure S1’. Crude 1H NMR spectrum of 2a in CDCl3) in the supporting information of the revised manuscript as suggested by reviewer-1
- The NMR spectra included in the supporting information should include a brief statement of the procedure use for characterization indicating the purification process.
Response:
hank you for pointing this out. The reviewer's comment is accepted; and we have included a brief statement of the procedure use for characterization and purification process as suggested by reviewer-1
Reviewer 2 Report
„Catalytic Acceptorless Dehydrogenation (CAD) of Secondary Benzylic Alcohols into Value-added Ketones using Pd(II)-NHC Complexes”
The manuscript by Mostafa et al. deals with the acceptorless dehydrogenation of secondary alcohols to produce the respective ketones. The authors demonstrate six Pd(II) NHC Complexes A-F as suitable catalysts for the acceptorless dehydrogenation of sec. alcohols with the proposed formation of elemental hydrogen, although the latter was not proven experimentally. The topic of this paper is certainly of interest for a broader community and of particular importance in the area of green and sustainable chemistry and can be accepted in the journal “Molecules”.
However, prior to publication the paper requires major revision and amendments as stated below.
1) This reviewer is confused about the identity of complexes A-F. The stated reference [59], according to which the Pd(II) complexes were prepared deals with ruthenium complexes. Palladium complexes are not mentioned in this reference at all. In the supporting information no NMR spectra, elemental analysis or crystal structures of A-F are given. If complexes A-F are literature reported, the please properly cite them or fully characterize new compounds.
2) I am wondering if the hydrogen formation was proven experimentally. Also related to this question: Have the authors investigated the reversibility of the reaction? Does the product ketone react to give the sec. alcohol under hydrogen atmosphere.
3) Table 1: For better comparison the authors should repeat entry 13 at 100°C. Explain the values in the parentheses, e.g. KOH (20). In the footnotes of Table 1 use “mmol” and not “mmols”. Please, also explain the abbreviations of the solvents. In some cases it may be difficult for readers to grasp what DCE or TFT etc. is.
4) Page 4. The paragraph following the table starts with two sentences “We looked into …” Please improve.
5) When addressing Figures and Schemes throughout the manuscript, please use terms like “Figure 1” or “Scheme 2” in normal (and not bold) letters. It appears as a mix as it stands now.
6) The list of references needs improvement. Reference [1] is missing. Reference [46] should state Pd(II) in the paper title. Figure 1 states complexes by S. Schneider and coworkers, but also this reference is missing.

Author Response
Manuscript ID: molecules-2433079R1 Date: June, 10th, 2023
Response to Reviewers
Dear Editor,
Thank you for giving us the opportunity to submit a revised version of draft of the manuscript entitled “Catalytic Acceptorless Dehydrogenation (CAD) of Secondary Benzylic Alcohols into Value-added Ketones using Pd(II)-NHC Complexes” by Professor Saleh et al. to be considered for publication as a research article in your prestigious journal of molecules - MDPI. We appreciate the time and effort that you and the reviewers dedicated to providing feedback on our manuscript and are grateful for the insightful comments on and valuable improvements to our paper. We have incorporated most of the suggestions made by the reviewers. Those changes are highlighted within the revised manuscript. Please see below, in blue, for a point-by-point response to the reviewers’ comments and concerns. All the reference has been rechecked and revised as suggested by the reviewers.
Reviewer’s comments & responses
Reviewer 2 comments
Recommendation: major revision.
“Catalytic Acceptorless Dehydrogenation (CAD) of Secondary Benzylic Alcohols into Value-added Ketones using Pd(II)-NHC Complexes”
The manuscript by Mostafa et al. deals with the acceptorless dehydrogenation of secondary alcohols to produce the respective ketones. The authors demonstrate six Pd(II) NHC Complexes A-F as suitable catalysts for the acceptorless dehydrogenation of sec. alcohols with the proposed formation of elemental hydrogen, although the latter was not proven experimentally. The topic of this paper is certainly of interest for a broader community and of particular importance in the area of green and sustainable chemistry and can be accepted in the journal “Molecules”.
However, prior to publication the paper requires major revision and amendments as stated below.
- This reviewer is confused about the identity of complexes A-F. The stated reference [59], according to which the Pd(II) complexes were prepared deals with ruthenium complexes. Palladium complexes are not mentioned in this reference at all. In the supporting information no NMR spectra, elemental analysis or crystal structures of A-Fare given. If complexes A-F are literature reported, the please properly cite them or fully characterize new compounds.
Response: Thank you so much for pointing this out. now we have revised the reference [59].
- I am wondering if the hydrogen formation was proven experimentally. Also related to this question: Have the authors investigated the reversibility of the reaction? Does the product ketone react to give the sec. alcohol under hydrogen atmosphere?
Response: The reviewer's comment is accepted; and here the hydrogen formation was proved experimentally and the explanation was added in the revised manuscript (Figure S1’’ GC traces after 24 hours reaction to confirm H2 evolution) (Experiments with intermolecular hydrogen transfer were carried out to verify H2 evolution in the reactions).
- Table 1: For better comparison the authors should repeat entry 13 at 100°C. Explain the values in the parentheses, e.g. KOH (20). In the footnotes of Table 1 use “mmol” and not “mmols”. Please, also explain the abbreviations of the solvents. In some cases it may be difficult for readers to grasp what DCE or TFT etc. is.
Response: Thank you for pointing this out. The reviewer's comment is accepted, and we have revised the entry 13 at 100°C. The values in the parentheses, e.g. KOH (20) means 20 mol% of KOH base. The abbreviations of the solvents have been added in the revised manuscript.
- Page 4. The paragraph following the table starts with two sentences “We looked into …” Please improve.
Response: The reviewer's comment is taken into consideration, and as a result, we have revised both the sentences.
- When addressing Figures and Schemes throughout the manuscript, please use terms like “Figure 1” or “Scheme 2” in normal (and not bold) letters. It appears as a mix as it stands now.
Response: Thank you for pointing this out. The reviewer is correct, and we have revised the captions like “Figure 1” or “Scheme 2” in normal letters as suggested by the reviewer-2.
- The list of references needs improvement. Reference [1] is missing. Reference [46] should state Pd(II) in the paper title. Figure 1 states complexes by S. Schneider and coworkers, but also this reference is missing.
Response: The reviewer's comment is taken into consideration, and as a result, here we have revised all the references carefully.
Round 2
Reviewer 1 Report
I would like to thank the authors for considering my recommedations and in my opinion the manuscript has improved considerably. However, I can not accept the manuscript in its actual form because there is a substantial limitation in the reaction mechanism. The authors still claim a metal-ligand cooperation mechanism. In my opinion there is no evidence of such mechanism but even more important is that the process described in Scheme 3 is a traditional inner-sphere mechanism. A metal-ligan mechanism should consider the participation of the ligand. I do not see any protonation or any change of the palladim ligands. My recomendation is just to rewrite the section of the mechanim.
Author Response
Recommendation: minor revision.
I would like to thank the authors for considering my recommendations and in my opinion the manuscript has improved considerably. However, I can not accept the manuscript in its actual form because there is a substantial limitation in the reaction mechanism. The authors still claim a metal-ligand cooperation mechanism. In my opinion there is no evidence of such mechanism but even more important is that the process described in Scheme 3 is a traditional inner-sphere mechanism. A metal-ligan mechanism should consider the participation of the ligand. I do not see any protonation or any change of the palladim ligands. My recomendation is just to rewrite the section of the mechanim.
Response: Thank you very much for your valuable comments, we revise the mechanism according to your esteemed opinion.
Reviewer 2 Report
The authors have addressed all my issues from the previous version. The manuscript can be accepted as it stands.
Author Response
Thank you very much for your acceptance